# Ultrafine Grain Ferrite Transformed from Fine Austenite Grains Produced by Dynamic Reversal Transformation

**DOI:** 10.3390/ma15248727

**Published:** 2022-12-07

**Authors:** Hongbin Li, Xiaoping Zheng, Lifeng Fan, Haiwei Xu, Yaqiang Tian, Xin Dai, Liansheng Chen

**Affiliations:** 1Key Laboratory of the Ministry of Education for Modern Metallurgy Technology, Tangshan 063210, China; 2School of Materials Science and Engineering, Inner Mongolia University of Technology, Hohhot 010000, China; 3Technology Center, Shougang Jingtang United Iron & Steel Co., Ltd., Tangshan 063200, China

**Keywords:** austenite refinement, dynamic reversal transformation, warm deformation, ultra-fine ferrite grains, texture

## Abstract

The medium carbon steel warm deformation was carried out in a Gleeble-3500 simulator, and the microstructure was observed on a scan electron microscopy (SEM) and optical microscope (OM). The results show that the dynamic reversal transformation (DRT) of austenite occurred during the multipass deformation at a temperature of 675 °C. The austenite grain size is about 3.4 μm at the stain of 2.67. The thermodynamics was discussed based on the stress activation model. The critical stress of DRT is in the range of 265.94–294.28 MPa, which is related to the Schmit factor, without considering the distortion energy. Meanwhile, the submicron ferrite was obtained after the air cooling stage. The texture of the ultrafine ferrite possessed the characteristics of good, deep drawing properties.

## 1. Introduction

Grain refinement has attracted many researchers’ interest and a lot of investigation on its use for improving strength and ductility simultaneously, with simple chemical composition and without additional heat treatment [1]. Therefore, many researchers investigated the methods to refine ferrite grains to obtain super mechanical properties without high alloy elements. The methods of producing ultrafine grained ferrite can be divided into two main categories: sever plastic deformation (SPD) and the advanced thermomechanical processes (ATP). SPD technologies [2] include several methods, such as the equal channel angular pressing, multiple large strain deformation, accumulative roll bonding [3], and high-pressure torsion [4]. ATP technologies include methods such as recrystallization of austenite during hot deformation [5], strain-induced ferrite transformation [6], intercritical hot rolling [7], dynamic recrystallization of ferrite during warm deformation, pronounced recovery of ferrite during warm deformation, and annealing and cold rolling and annealing of martensitic steel. The differences between the above two categories are obvious. For the SPD methods, a well-designed strain path is more important and also more feasible. It needs considerable ingenuity and investment for application because of the high strain and complex strain path. However, the strain of ATP technology is about 1.0–3.6, which is lower than that of SPD technology. Meanwhile, the strain route of ATP is far easier than that of SPD technology. Consequently, the potential possibility for industrial manufacture under the present production condition of ATP technology is more than that of SPD technology.

Therefore, in iron & steel productive processing, ATP is adopted usually. The method of recrystallization of austenite during hot deformation has been adopted widely. Meanwhile, based on it, the deformation of unrecrystallization of austenite was developed to refine ferrite by transformation from austenite to ferrite due to the increase of the rate of new phase nucleation. Based on real production conditions, the alloying was considered to realize the controlling rolling and controlling cooling in order to refine the size of the ferrite grain size of finished production. It is obvious that the alloying would increase the production difficulty and cost in iron & steel production. Therefore, the refinement of ferrite grain size is an effective way to elevate the mechanical properties without alloying. However, the size of ferrite was about 1μm, obtained by the strain-induced ferrite transformation, due to the high transformation temperature and low cooling rate required because of the large cross size of the iron and steel production. It caused the size of new ferrite grain transited from the austenite to increase. The growth of new ferrite grain decreased the refinement results of the strain-induced ferrite transformation.

Based on the above discussion and current production conditions, the refinement of austenite may be an effective method to accomplish ferrite grain size refinement of final products, such as bars, plates, strips, and wire rods.

According to the results of [8], refining austenite grain size is an effective method to produce ultrafine ferrite grain. Yao et al. [9] and Sun et al. [10] proposed that ferrite size will be refined to the level of the sub-micron, and the cementite precipitated in the form of particles with nano scale on the matrix of fine ferrite, if the size of the austenite grain was refined to the level of 3–4 μm. Yet, how to refine the austenite grain size to the level of 3–4 μm is a great challenge based on normal austenization deformation conditions.

The results of [8] indicated that the deformed ferrite grains might transform into fine austenite grains (mean grain size is about 5 μm) during the rapid rate heating process used. This indicates that distortion energy contributes to austenite grain refinement during the reversal transformation. The reversal transformation from ferrite to austenization is an effective way to obtain fine austenite grains, which is consistent with the results of [11]. The grain size of austenite could be refined by the reversal transformation and achieve the level of 5 μm. However, this needs two required conditions: the distortion energy in the matrix of the ferrite and a high heating rate. Yet, these conditions are difficult to meet based on current production conditions.

It is well known that the dynamic transformation from austenite to ferrite can refine the ferrite grain size [12]. In this way, the dynamic reversal transformation (DRT) may be an effective way to refine austenite grains to 3–4 μm. Additionally, ferrite deformed at the temperature near the A_1_ with high strain rate may allow the dynamic reversal transformation to occur, which is caused by the distortion energy and the increased temperature due to the heat transited from plastic work, thus achieving the synergistic action of increasing temperature and deformation. It is fortunate that the temperature increases due to the transformation from plastic work to heat during the finish rolling stage of hot rolled strip production with high strain rate. However, there are two questions that should be clarified. First, does the DRT occur or not during the finish rolling stage, and second, what size and texture of ferrite grain is obtained and transferred from reversal austenite during the subsequent air cooling.

Therefore, this research aims to check the DRT occurrence and the characteristics of ferrite grains transferred from reversal austenite. Meanwhile, the occurrence possibility and thermodynamics of DRT are discussed.

## 2. Materials and Methods

The experimental material is a type of C-Mn steel with the chemical composition 0.46 C, 0.23 Si, 0.72 Mn, 0.03 P, 0.03 S,0.02 Cr, 0.02 Ni, 0.03 Cu, Fe: Bal., (wt%).The specimen size is 10 × 15 × 20 mm, with the microstructure composed of ferrite and pearlite. The grain size of ferrite in the specimens is about 20 μm. The deformation tests were conducted on a Gleeble-3500 thermal mechanical simulator (Dynamic Systems Inc., Austin, TX, USA).The deformation temperature is 650 °C. The test schemes are shown in Figure 1a.The strain path is proposed in Table 1. In order to measure the yield stress of the undercooled austenite, the cylinder sample was compressed to the true strain of 1.16 in a single pass. The size of the cylinder sample is φ8 × 12 mm. The scheme is shown in Figure 1b.

After the 6 passes compression, the plane strain specimens were quenched in water immediately. The microstructure was observed by SEM(scanning electron microscope, Nova NanoSEM430, FEI, Boston, MA, USA) and OM(optical microscope, DMI5000M 03040121, Germany). The center part of deformed specimens was observed by SEM after mechanical polishing and erosion in 4% Nital 30 s. To observe the reversal transformation austenite boundaries, the samples were etched in the mix solution of picric acid, hydrochloric acid erosion 5–10 min at the temperature of 55 ± 5 °C. The size of the austenite was measured by OM. The austenite grain size was calculated by the quantitative metallography method. After the 6 passes deformation and cooling to room temperature in air, the specimens for electron back-scattered diffraction (EBSD, Oxford NORDLYS 2S, Oxford, UK)analysis were cut, mechanically grounded 2500 # and electrolytically polished. The scan step of EBSD detection is 0.15 μm. The misorientation of grains between 2° and 15° is defined as low-angle grain boundaries (LAGBs), and the misorientation above 15° is defined as high-angle grain boundaries (HAGBs).

## 3. Results

### 3.1. Variation of Follow Stress and Temperature

The flow stress and temperature variation curves are shown in Figure 2. It is obvious that the follow stress peaks of the first two passes are much higher than those of the following passes, which are not much different. Furthermore, the peak temperature of the first three passes is higher than that of the following passes. According to the variation of temperature during the multi-pass deformation processing, the deformation process is divided into two stages. The two stages are divided by the red dotted line as shown in Figure 2.

During the first stage, the trend of the temperature is increasing. At the second pass, the temperature reached the maximum of about 695 °C. During the second stage, the temperature variation was small. The range of temperature variation was approximately 670 to 675 °C, and the follow stress increased at the initial strain of every pass. In addition, during the intervals, the stress decreased rapidly. The temperature increased during the interval of the first and the second passes, but during the interval of the second and third passes, the temperature decreased, and the follow stress decreased sharply, as well. During the third and fourth passes, the temperature and follow stress decreased sharply. However, during the second stage, the peak stress of the fourth and fifth passes was almost the same, and the temperature variation was small, although it was at the intervals between the fourth and fifth passes. Due to the low strain of the last pass, the temperature decreased.

### 3.2. Variation of Microstructure during Multi Passes Compression

After the six passes of plane strain compression, the specimens were quenched in room temperature water immediately. The morphology of the final microstructure is shown in Figure 3. The microstructure is composed of matensite with no ferrite found, indicating that the reversal transformation is complete after six passes.

This indicates that the reversal transformation from ferrite to austenite occurred during the six passes plane strain compression. It testified the hypothesis of the occurrence of the reversal transformation from ferrite to austenite during the multi-pass high strain rate compression. The reversal transformation austenite formation during the compression is characterized as shown in Figure 4.

It is obvious that the content of reversal austenite increased with the compression passes. After the third pass of compression, the fraction of reversal austenite was low and approximately one third in the field of vision, as shown in Figure 4a. Meanwhile, the size of new reversal austenite was approximately~4 μm, which was close to the results reported by Refs. [9,10].With the compression processing, the fraction of residual ferrite decreased sharply, and the fraction of reversal austenite increased to about a half of the total, as shown in Figure 4b. After the fifth transformation, the reversal transformation processing was almost complete, except for seldom residual ferrite, as shown in Figure 4c. After the sixth compression, the reversal transformation processing was almost complete, and there was no residual ferrite found in the field of vision, as shown in Figure 4d, which was consistent with the result in Figure 3. The average grain size of the austenite was about 3.4 μm, which was the adequate size required by Refs. [9,10]. Based on the results shown in Figure 4, it could be concluded that the reversal transformation might start at the intervals of the second and third passes. This showed that the rate of reversal transformation was high during the multi-pass compression, and it was controlled by the deformation strain. Meanwhile, the intervals played a key role for the austenite reversal transformation.

### 3.3. Variation of Microstructure during Air Cooling after Multi-Pass Compression

The fine austenite could transform into the ferrite and pearlite during the following air cooling procedure. The morphology of the product of transformation from austenite was observed by EBSD, TEM. The size of the ferrite grains transformed from austenite during the air cooling procedure was about 0.62 μm, as shown in Figure 5a. The misorientation of the fine ferrite grains is shown in Figure 5b. The average misorientation was 22.5°. This indicates that most of the misorientation angle between the fine ferrite grains was large angle. The IPF map of the ferrite is characterized in Figure 5c. It is obvious that some fine ferrite with similar orientation was arranged in a line, and it showed strong texture characteristics. The Polefigure of the (100) crystal plane and ODF of φ = 45° are shown in Figure 5e,f. It shows a strong texture mainly composed of (223)<142>, (111)<112>, (100)<110> and (332)<5-73>. The fraction of main texture is shown in Table 2. As shown in Figure 5d, the morphology of most of the pearlite is granular and the size is about 100 nm. These particles distributed intragranular and on the grain boundaries. Meanwhile, there are some pearlites, which are like short rods, but the fraction of short rods like pearlite is much lower than that of the granular pearlite. The difference of the pearlite’s morphology might be attributed to the formation mechanism. It is obvious that there was no pearlite found in Figure 3. Based on the fact discussed above, the short rod-like pearlite was transformed from the pearlite lamella residual, which was such a small amount that it cannot be found in Figure 3.

## 4. Discussions

### 4.1. Analysis of Multi-Pass Compression

During the multi-pass compression, the deformation compression could be divided into two stages, as shown in Figure 2. During the first stag, the temperature increased, owing to the plastic work transformation in heat in the high strain rate deformation [13]. At the interval of the first and second passes, the temperature increases sharply and the flow stress decreases sharply. The decrease of flow stress can be attributed to the decrease of dislocation due to the static recovery, which decreases the distortion energy caused by the first pass compression. The released distorted energy transformed into the heat with the decreasing of dislocation density. Due to the limitation of the interval time, the heat cannot be released into the environment in time, which results in the temperature increasing sharply during the interval time. But at the interval of the second and third passes, the temperature decreased first and then increased. There seemed to be some phenomena absorbing the heat transited from the annihilation of dislocation. It might be attributed to the reversal transformation from distorted ferrite to austenite because the reversal transformation is the only heat absorbing phenomenon in this test. This indicated that the reversal transformation might have occurred at the interval of the second and third passes. The temperature increased and then decreased during the interval of the second and third passes, which might be caused by the fact that the quantity of heat from the dislocation annihilate was more than what the reversal transformation needed. This indicated that the amount of the reversal transformation was low. In addition, the follow stress decreased sharply after the peak stress, which might be attributed to the reversal transformation during the second pass compress, similar to dynamic recrystallization. However, this needs to be further researched. Meanwhile, at the end of the first stage (the intervals of the third and fourth passes), the temperature decreased sharply without increasing. This indicated that the amount of the reversal transformation was more than that during the previous interval.

During the second stage, the peak stresses were less than that of the first and second passes. The temperature was about 675 °C, and the increase of temperature is low during the intervals. This indicates that there may be some phenomenon that absorbed the heat transformed from plastic work, which is similar to that of the intervals after the second and third passes. It may be the austenite reversal transformation, too. To verify the hypothesis, the interrupted quenching test was carried out. The specimen was quenched after the third, fourth, and fifth passes and etched in the mix solution. If it is true that the austenite reversal transformation occurred, the austenite grains’ boundaries would appear and the austenite grain could be measured. In Figure 4, there are some austenite grains, just as shown as the identification in Figure 4. This indicates that the above hypothesis is true. By comparison to the content of austenite after the different passes of deformation, it is clear that the austenite content increases with the strain increasing. The size of the austenite grains formed during deformation is about 3.4 μm, as shown in Figure 4d.

The fine austenite grains with a size of 3 μm in Ref. [9] were produced by cycle quenching and austenitizing three times. The austenite grains with a size of 1–2 μm were prepared by powder sintering processing with nanometer powder, as in Ref. [10]. The size of austenite grains obtained in the present research is slightly more than that in Refs. [9,10], but the present process is much simpler than those reported in Refs. [9,10], and there is great potential for industrial application.

### 4.2. Thermodynamic Analysis of DRT

Based on the above research, it is obvious that the reversal transformation occurred during the deformation, and the austenite grain is less than the results of Ref. [8]. Generally, the austenite transformation occurred when the temperature exceeded the Ae_1_. But the Ae_1_ of test steel is about 726 °C, as calculated in Ref. [14], which is more than the deformation temperature. Thus, the comparison between driving force and obstacle energy should be analyzed.

Based on the thermodynamics of transformation, one of the driving forces is evaluated as the difference between the flow stress ($\sigma_{}$) of the ferrite stress up to the critical strain and the yield stress ($\sigma_{\gamma }$) of the fresh austenite that takes its place [15]. Another driving force is the distortion energy, which was in the form of dislocations. According to the results of Ref. [16], the distortion energy only provides a small part of the driving force, and it can be neglected.

The obstacle energy is composed of free energy ($\Delta G_{\alpha -\gamma }$), shear accommodation (${\left(W/V\right)}_{SA}$), and dilatation work (${\left(W/V\right)}_{D}$) [17].

If the reversal transformation occurred, the difference of ferrite follow stress and austenite yield stress meets Equation (1), without considering the distortion energy, and the ferrite follow stress should be more than the $\sigma_{c}$ (critical stress for austenite DRT).
(1)$$\sigma_{c}-\sigma_{\gamma }=\Delta G_{\alpha -\gamma }+{\left(W/V\right)}_{SA}+{\left(W/V\right)}_{D}$$

Meanwhile, the shear accommodation work and dilatation work can be expressed as Equations (2) and (3).
(2)$${\left(W/V\right)}_{SA}=m\times \sigma_{c}\times 0.36$$
(3)$${\left(W/V\right)}_{D}=\lambda \times \sigma_{c}\times 0.03$$

Therefore,
(4)$$\sigma_{c}=(\Delta G_{\alpha -\gamma }+\sigma_{\gamma })/(1+0.36m+0.03\lambda )$$
m and λ is the Schmid factor and shear strain between the austenite and ferrite, respectively. $\sigma_{c}$ is the critical stress of the ferrite transformed to austenite, and $\lambda =\sqrt{m}$ [15]. The Schmid factor is from 0.2 to 0.5, and 1 MPa=7.2 J/mol [12,15]. The undercooled austenite yield stress at 680 °C is about 200 MPa, according to the flow stress of undercooled austenite, as shown in Figure 6a. The value of $\Delta G_{\alpha -\gamma }$ is about 0.9 kJ/mol, as calculated by the software of the Fact Sage databases FTPS and FTSTEL, as shown in Figure 6b. Therefore, the range of $\sigma_{c}$ is from 265.94 to 294.28 MPa, which is related to the Schmid factor.

It is obvious that the highest stress of ferrite is more than that of the critical value during the multi-pass deformation, as shown in Figure 2. The critical condition of the reversal transformation is achieved, which means that the thermodynamic condition is met.

Another important parameter is the strain at which the DRT occurred. Based on the critical stress and the multi passes follow stress curve, the critical strain of DRT may be about 0.05, but the second pass stress decreases sharply. Meanwhile, the increase of second pass temperature is less than that of the first pass deformation, which may be attributed to the DRT. Therefore, the DRT critical strain and the mechanics of the follow stress abnormal variation of the second pass should be further researched in detail.

### 4.3. Analysis of Microstructure Transformation from Fine Austenite to Ferrite during Air Cooling

The DRT occurred during the multi-pass deformation, and the rate of DRT increased with the strain. After six passes, the DRT was completed, based on the results of Figure 4. The size of the ferrite transformed from fine austenite was detected by EBSD. The results are shown in Figure 5. It can be determined that the average size of the ferrite grain is about 0.64 μm, and the submicron ferrite is more than 90%. The average misorientation is 22.5° and the big angle boundaries are more than 70%.This indicates that the fine austenite grain could refine the ferrite grain to the level of the submicrometer, and the size of the carbide particles precipitated during the transformation is about 50 nm because of the divorce eutectoid transformation during the air cooling [18,19], as shown in Figure 5d. Due to the low transformation temperature (<680 °C), the coarsening of new ferrite grains was restrained, which is one of the main reasons why the submicrometer ferrite grains could be preserved at room temperature. The results of Ref. [20] obtained ultrafine ferrite with 1.1 μm by the strain induced transformation from austenite with the true strain of 1.0. Meanwhile, Okistuet et al. [21] prepared fine ferrite grains with the size of 0.49–0.85 μm by cold rolled and annealing processing, and the plastic strain was about 2.8.

Moreover, the nanoscale ferrite grains were obtained by severe plastic deformation (SPD), with the true strain about 4.0 [22], which obviously were difficult to achieve under the current production conditions. In this research, the strain is less than that of SPD, and the scale of ferrite grains is less than that of Ref. [20]. Hence, this shows the advantage of refining the ferrite grains with lower strain, and the great potential for the industrialized application.

For structure materials, the mechanical properties of iron and steel production were not determined only by the size of the ferrite grains. The texture of the ferrite was another important factor, especially for the deep drawing part in the vehicle manufacture processing. But the variation of the ferrite texture was affected by many factors, such as the austenite recrystallization, the reduction of the austenite during the temperature range of unrecrystallization, the transformation, and the deformation of the ferrite at high temperature [23,24]. But in this research, the factor of austenite recrystallization could be neglected due to the low deformation temperature and the high strain rate; thus, the main affected factors were the others. Therefore, the texture of ultra-ferrite grains was discussed.

To reveal the texture in detail, the pole figure of (100) and ODF were established, as shown in Figure 5e,f. It is distinct that the texture of ultra-ferrite grains was mainly comprised of (100)<110>, (223)<142>, (111)<112> and (332)<5-73>, just as shown in Table 2.

The texture of (332) was transformed from the Bs texture (110) of austenite during the phase transformation, and the Bs texture was formed during the no-recrystallization temperature range [25]. In this research, the reversal austenite formed in the earlier stage would deform during the deformation, and the recrystallization could not occur due to the low temperature and high strain rate. Therefore, the deformed reversal austenite with high Bs texture transformed to the ferrite with the texture of (332). Based on Table 2, the fraction of (332) was 14.1%, which indicated that the Bs texture (110) was the main texture of the fine austenite deformed during the multi-pass deformation.

Furthermore, the reversal austenite formed during the later period of stage 2 might show the cube texture of (100)<001> and transform to the ferrite with the rotated texture of (100)<110>.This could be inherited from the recrystallization austenite grains during the hot rolling processing, based on the K-S(Kурдюмoв-Sachs) relationship [26].

As the results show in Ref. [27], the texture (223) formed from ferrite deformation during stage 2 [28] could be transited to the texture of (111) and (554), which has the benefit of good, deep drawability. In addition, the ferrite texture of (332) could shift into the texture of (554) then to (111) during the next ferrite cold rolling processing. Based on the above test results and discussion, it is obvious that the ultrafine ferrite may have the potential to possess the good, deep drawing property due to the ferrite texture characteristic.

## 5. Conclusions

Austenite DRT occurred during medium carbon steel warm deformation at 650 °C with high strain rate multi-pass compression.Fine austenite grains were obtained from the DRT, and the size of the austenite grains was about 3.4 μm at the total strain 2.67.Ultrafine ferrite grain was obtained and the size was approximately 0.62 μm. Additionally, the ferrite texture was characterized by good, deep drawing properties.

## Figures and Tables

**Figure 1 materials-15-08727-f001:**
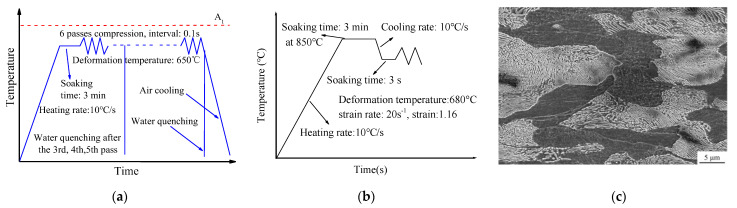
Deformation test scheme; (**a**) Multi-pass plane strain, (**b**) Undercooled austenite deformation, (**c**) Initial microstructure of the specimen.

**Figure 2 materials-15-08727-f002:**
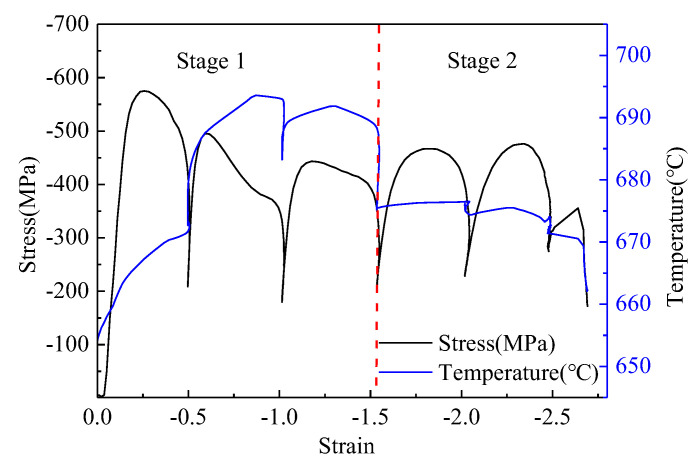
Temperature and flow stress curves variation during multi-pass deformation.

**Figure 3 materials-15-08727-f003:**
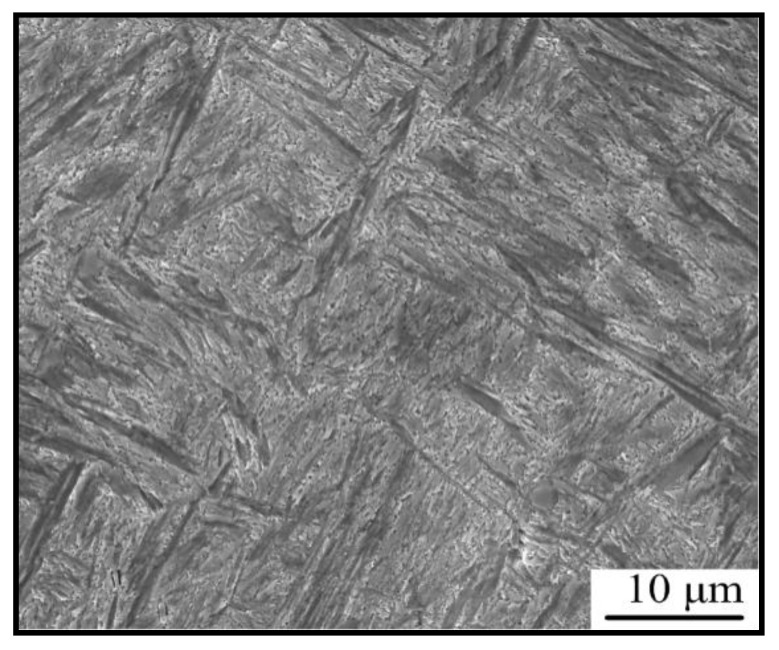
Microstructure after six passes deformation after quenched in water.

**Figure 4 materials-15-08727-f004:**
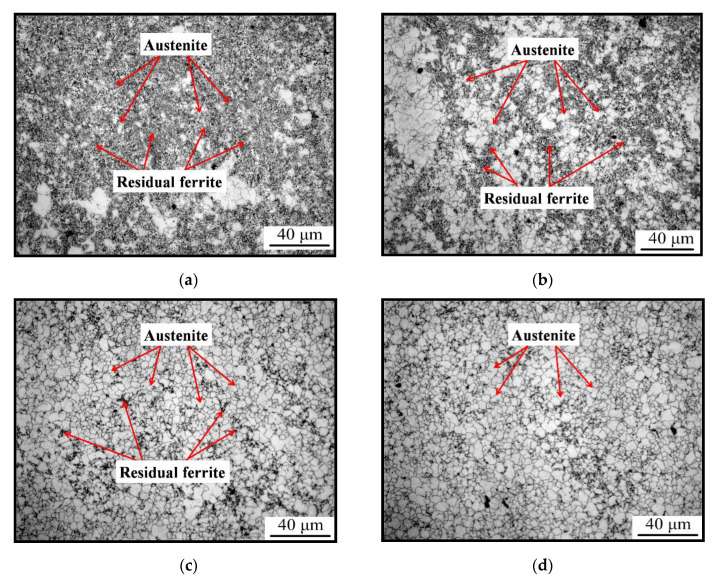
Austenite grain after different passes: (**a**) after 3rd pass, (**b**) after 4th pass. (**c**) after 5th pass, (**d**) after 6th pass.

**Figure 5 materials-15-08727-f005:**
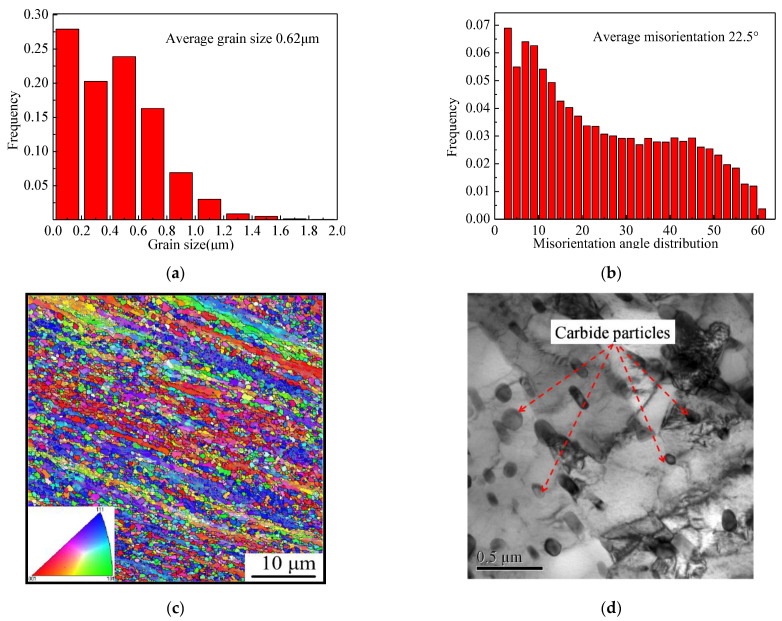
Characteristics of ferrite and carbide.(**a**) Distribution of grain size, (**b**) Distribution of misorientation angle, (**c**) IPF map, (**d**) Carbide morphology, (**e**) Polefigure of (100) crystal plane, (**f**) ODF of φ = 45°.

**Figure 6 materials-15-08727-f006:**
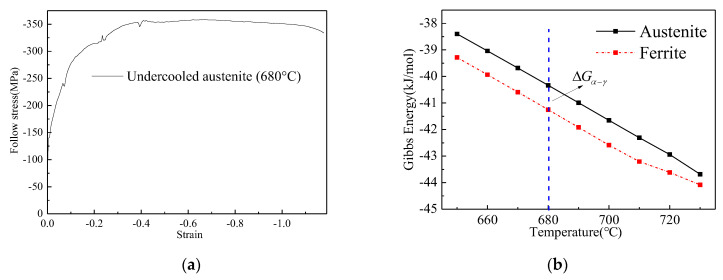
Undercooled austenite follow stress and Gibbs energy of ferrite and austenite; (**a**) austenite follow stress of 680 °C, (**b**) Gibbs energy of ferrite and austenite.

**Table 1 materials-15-08727-t001:** Multi passes compression strain path of plane strain tests.

Pass	True Strain	Strain Rate (s^−1^)	Interval (s)
1	0.39	20.00	0.1
2	0.53	20.00	0.1
3	0.52	20.00	0.1
4	0.49	20.00	0.1
5	0.47	20.00	0.1
6	0.27	20.00	--

**Table 2 materials-15-08727-t002:** Fraction of each texture.

Texture	(223)<142>	(111)<112>	(100)<110>	(332)<5-73>
Content	17.1%	9.47%	5.31%	14.1%

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
