# Peer review of "Ultrafine Grain Ferrite Transformed from Fine Austenite Grains Produced by Dynamic Reversal Transformation"

_materials, 2022, doi:10.3390/ma15248727_

Round 1
Reviewer 1 Report
Ultrafine grain ferrite transformed from fine austenite grains produced by dynamic reversal transformation
The paper is well organized and can be accepted after major revision.
1) The aim of the present work is missing. The authors need to rewrite it.
2) The deformation temperature in the text is missing.
3) The 100-pole figure was incomplete. The arrows 100, 111, 332, 223, etc are not correct.
4) What are the chemical compositions of carbide particles?
Author Response
1) The aim of the present work is missing. The authors need to rewrite it.
Thanks very much, the aim of the present work was added in the instruction section, and highlighted.
2) The deformation temperature in the text is missing.
Thanks. The deformation temperature was added in the section 2 Materials and methods, and highlighted.
3) The 100-pole figure was incomplete. The arrows 100, 111, 332, 223, etc are not correct.
Thanks, the 100-pole figure has been completed and the arrows were corrected.
4) What are the chemical compositions of carbide particles?
The carbide particles should be Fe3C, so the chemical compositions are Fe and C.

Reviewer 2 Report
The article is devoted to the study of the possibility of the medium carbon steel austenite grain refinement using the dynamic reversal transformation of austenite during the multipass warm deformation. The steel microstructure was investigated and the thermodynamic calculations were carried out based on the stress activation model.
Some remarks:
1. Abstract: “The results show that the dynamic reversal transformation (DRT) of austenite occurred during the multipass deformation at temperature of 675 °C.”. However, figure 1a indicates a deformation temperature of 650 °C.
2. Introduction: The sentence “It realized the synergistic action of temperature increasing and deformation.” repeats two times.
3. Where is the investigated steel used for? Specify, please.
4. Materials and methods: "The specimen size is 10×15×20mm with the microsturcture composed of ferrite and pealite." Must be pearlite.
5. Explain what the line Ae1 in Figure 1a means. Maybe you mean Ac1?
6. Why exactly 6 deformation passes were performed?
7. Please add information about research equipment (OM, SEM, TEM).
8. Please add more details about the initial state of the steel (figure).
9. 3.2. Variation of microstructure during multi passes compression: "And there is no ferrite found, it indicates that the reversal transformation is completely after 6 passes. It indicates that the reversal transformation from ferrite to austenite occurred, during the six passes plane strain compression". The sentences almost repeat each other.
10. In the caption to figure 4, information on the presence of ferrite grains should also be added.
11. In Figure 5e it is not clear what the arrows are pointing to.
12. 4.1. Analysis of multi passes compression: "The decrease of flow stress can be attributed to the decrease of dislocation decrease, due to the static recovery, which decreases the distortion energy caused by the first pass compression." Probably one word "decrease" is needless.
13. 4.3. Analysis of microstructure transformation from fine austenite to ferrite during the air cooling: “… which might be inherited from the recrystallization austenite grains during hot rolling processing, based on the K-S relationship…” Please clarify the abbreviation K-S.
14. References are in Roman numerals. Please change to Arabic numerals.
Author Response
1. Abstract: “The results show that the dynamic reversal transformation (DRT) of austenite occurred during the multipass deformation at temperature of 675 °C.”. However, figure 1a indicates a deformation temperature of 650 °C.
Thanks for the constructive suggestion, in the Fig. 1a, the 650°C was the setting deformation temperature. But during the deformation process, the temperature increased, so the real deformation became 675 °C.
2. Introduction: The sentence “It realized the synergistic action of temperature increasing and deformation.” repeats two times.
Thanks, the needless sentence has been deleted.
3. Where is the investigated steel used for? Specify, please.
Thanks, the steel will be used to manufacture some fittings for machine.
4. Materials and methods: "The specimen size is 10×15×20mm with the microsturcture composed of ferrite and pealite." Must be pearlite.
Thanks, It is our mistake, the “pealite” has been repeated by “pearlite”, and highlited.
5. Explain what the line Ae1 in Figure 1a means. Maybe you mean Ac1?
Thanks, The Ae1 should be A1. It has been modified to A1.
6. Why exactly 6 deformation passes were performed?
Thanks. This research was carried out with a background of strip finish rolling processing. The passes of finish rolling were about 5-7, generally. So the test was performed with 6 passes.
7. Please add information about research equipment (OM, SEM, TEM).
Thanks, the information of the research equipment have been added, and highlighted
8. Please add more details about the initial state of the steel (figure).
Thanks, the figure of the initial state of the steel was expressed in Figure 1
9. 3.2. Variation of microstructure during multi passes compression: "And there is no ferrite found, it indicates that the reversal transformation is completely after 6 passes. It indicates that the reversal transformation from ferrite to austenite occurred, during the six passes plane strain compression". The sentences almost repeat each other.
Thanks very much, the repeated sentences have been corrected.
10. In the caption to figure 4, information on the presence of ferrite grains should also be added.
Thanks. In figure 4, the main aim was to explore the progress of the DRT during deformation. So the corrosive liquid was the mix solution of picric acid, hydrochloric acid erosion. But this corrosive liquid can not characterize the ferrite grains. So the ferrite grains were fuzzy.
11. In Figure 5e it is not clear what the arrows are pointing to.
Thanks, the Figure 5e was dealt, and the arrows were moved to the suitable position.
12. 4.1. Analysis of multi passes compression: "The decrease of flow stress can be attributed to the decrease of dislocation decrease, due to the static recovery, which decreases the distortion energy caused by the first pass compression." Probably one word "decrease" is needless.
Thanks, it is my mistake. The needless “decrease” has been deleted.
13. 4.3. Analysis of microstructure transformation from fine austenite to ferrite during the air cooling: “… which might be inherited from the recrystallization austenite grains during hot rolling processing, based on the K-S relationship…” Please clarify the abbreviation K-S.
Thanks, the the abbreviation K-S has been clarified and highlighted.
14. References are in Roman numerals. Please change to Arabic numerals.
Thanks, the Roman numerals have been changed to Arabic numerals.

Round 2
Reviewer 1 Report
In the revision version still following points need to change-
1. " Therefore, this research arms to check the DRT". Should be Aims.
2. The following sentence needs to revise-
"The deformation tests were conducted on Gleeble-3500 thermal mechanical simulator (Dynamic Systems Inc., Austin, TX, USA). And the deformation temperature was 650 °C" .
Sentence construction is not correct.
Author Response
1. " Therefore, this research arms to check the DRT". Should be Aims.
Thanks, it has been revised and highlighted in the manuscript.
2. The following sentence needs to revise-"The deformation tests were conducted on Gleeble-3500 thermal mechanical simulator (Dynamic Systems Inc., Austin, TX, USA). And the deformation temperature was 650 °C" .Sentence construction is not correct.
Thanks, the sentence has been changed and highlighted in the manuscript. The corrected sentences are listed as follows.
The deformation tests were conducted on a Gleeble-3500 thermal mechanical simulator (Dynamic Systems Inc., Austin, TX, USA). The deformation temperature is 650 °C.
